# NEURAL COMPRESSION OF 3D MESHES USING SPARSE IMPLICIT REPRESENTATION

**Jianqiang Wang[†], Siyu Ren[†], and Junhui Hou[*]**
Department of Computer Science, City University of Hong Kong
`wang.jq@cityu.edu.hk, siyuren2-c@my.cityu.edu.hk,`
`jh.hou@cityu.edu.hk`

## ABSTRACT

The growing demand for high-quality 3D mesh models has fueled the need for efficient 3D mesh compression techniques. However, existing methods often exhibit suboptimal compression performance due to the inefficient representation of mesh data. To address this issue, we propose a novel neural mesh compression method based on Sparse Implicit Representation (SIR). Specifically, SIR records signed distance field (SDF) values only on regular grids near the surface, enabling high-resolution structured representation of arbitrary geometric data with a significantly lower memory cost, while still supporting precise surface recovery. Building on this representation, we construct a lightweight Sparse Neural Compression (SNC) network to extract compact embedded features from the SIR and encode them into a bitstream. Extensive experiments and ablation studies demonstrate that our method outperforms state-of-the-art mesh and point cloud compression approaches in both compression performance and computational efficiency across a variety of mesh models. The source code is available at https://github.com/yydlmzyz1/SIR-SNC.

## 1 INTRODUCTION

With the rapid advancement of applications such as virtual reality, robotics, and autonomous driving, the demand for more detailed and diverse 3D mesh models is growing rapidly. However, due to inherent limitations in network bandwidth, the development of efficient and robust 3D mesh compression techniques has become increasingly urgent. Existing mesh compression methods (Maglo et al., 2015), such as Draco (Google, 2025) and Video-based Dynamic Mesh Coding (V-DMC) (MPEG, 2025; Zou et al., 2025), all compress mesh vertices and their connectivity directly, often leading to substantial geometric distortion at low bitrates. Point cloud compression (PCC) techniques (Schwarz et al., 2019) offer an alternative to 3D mesh compression according to the transformation between point clouds and meshes, but there are severe distortions during the transformation between these two representation formats due to the discontinuous and unstructured nature of point clouds.

The suboptimal performance of existing representative geometry compression methods is primarily constrained by the inherently irregular structure of the explicit mesh representations themselves. Recently, widely used implicit fields, such as the signed distance field (SDF), can convert irregular meshes into tensors by sampling uniformly distributed grids in 3D space. However, such a representation suffers from redundant dense sampling, leading to an explosive growth of data volume as resolution increases. Since only sparse grid cells near the surface carry critical information, while most distant cells are useless, a more efficient representation of the implicit distance field is imperative. Motivated by this insight, we propose a Sparse Implicit Representation (SIR) that leverages the strengths of implicit distance fields, which offer continuous geometric precision, and sparse representations, which provide data efficiency, for compact, accurate mesh surface representation. We further propose a lightweight Sparse Neural Compression (SNC) module to efficiently extract the sparse latent features and encode them into the bitstream.

Fig. 1 illustrates the proposed compression pipeline. The original mesh is first converted into a sparse SDF tensor. Then, in the compression stage, the sparse SDF tensor is embedded into a compact

---

[*]Corresponding author. [†]Equal contribution.

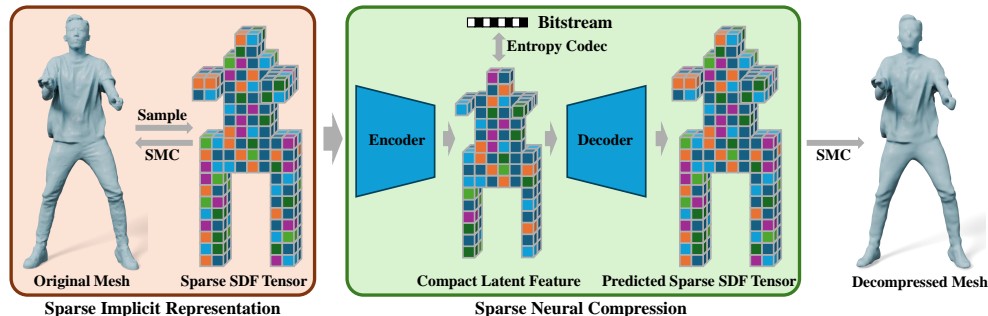

Figure 1: The pipeline of proposed SIR-based neural mesh coding. It first represents irregular 3D meshes into a regular Sparse SDF Tensor, a new implicit representation, and then an auto-encoder network is utilized to compress these tensors, obtaining Compact Sparse Features of each mesh, which is then converted into bitstreams through an entropy codec.

latent feature with lower resolution through a sparse convolutional autoencoder (AE) network. The feature is quantized and encoded with entropy into the bitstream. The encoder and decoder of AE correspond to the compression and decompression processes, respectively.

We perform extensive experiments across diverse datasets, demonstrating that the proposed neural mesh coding framework based on SIR and SNC achieves significant improvements over state-of-the-art alternatives, including mesh coding algorithms Draco (Google, 2025) and V-DMC (MPEG, 2025; Zou et al., 2025); PCC methods G-PCC (MPEG, 2025) and SparsePCGC (Wang et al., 2023); and another SDF-based approach, NeCGS (Ren et al., 2025). Furthermore, our method exhibits fast encoding and decoding speeds, a compact model size of only several hundred kilobytes (KB), and robust performance across different types of meshes.

We summarize our main contributions as follows.

- We propose sparse implicit representation (SIR), enabling structured and accurate representation for 3D meshes, while achieving computationally and memory-efficient processing through sparse operations.

- We further develop a sparse neural compression (SNC) method leveraging SIR, which demonstrates substantial improvements over state-of-the-art mesh compression methods while maintaining a fast encoding/decoding speed and a lightweight model size.

- We offer a *fresh* perspective to 3D mesh compression and elevate the compression performance to a new level, with extensive experiments across diverse mesh models for validation.

## 2 RELATED WORK

3D mesh is among the most common 3D geometry representations, yet geometry representation extends beyond this explicit format. In this section, we review representative and recent methods in 3D representation and compression, focusing on the accurate and efficient modeling and compression of geometric surfaces.

**Implicit Geometry Representations.** Different from widely used explicit representations, such as point clouds and triangle meshes, where the surfaces are represented as sampled points or triangles, implicit representations characterize surfaces via isosurfaces of continuous fields. Binary Occupancy Fields (BOFs) (Kazhdan & Hoppe, 2013; Mescheder et al., 2019) and Signed Distance Fields (SDFs) (Park et al., 2019; Atzmon & Lipman, 2020) are two most widely used implicit fields and the surface can be easily extracted from them through Marching Cubes (WE, 1987). However, BOF and SDF divide the whole space into two parts, inside and outside the surface, and they can only represent watertight meshes, limiting their applications to more general meshes. Unsigned Distance Fields (UDFs) (Chibane et al., 2020; Ren et al., 2023) can represent more general meshes; however, it is extremely challenging to extract a surface from them.

**3D Mesh Compression.** Extensive research on mesh compression has been conducted since the 1990s, with comprehensive reviews available in (Peng et al., 2005; Maglo et al., 2015). Mesh com-

pression typically involves both vertex coordinates and topological connectivity compression. The widely adopted Draco (Google, 2025) library orders vertices using Morton codes for predictive coding and employs the EdgeBreaker algorithm (Rossignac, 2002) to compress the topological relationships. In the developing MPEG mesh coding standard V-DMC (MPEG, 2025; Zou et al., 2025), the original mesh is decomposed into a base mesh and a set of displacement vectors via subdivision and deformation for separate compression, achieving superior performance. Additionally, some alternative approaches convert the mesh to 2D geometry images (Gu et al., 2002; Hou et al., 2014a;b; Zhang et al., 2022; 2023; Zeng et al., 2024), where 2D image/video encoders are applied for compression. These mesh coding approaches relying on explicit topological representations often suffer from serious quantization distortion and topological breakage in complex geometries, leading to a diminishing compression efficiency at higher compression ratios.

**3D Point Cloud Compression.** An alternative approach to compressing 3D meshes involves first converting them into point clouds and subsequently applying point cloud compression (PCC) techniques. PCC encompasses a diverse range of methods leveraging various representations, including 3D voxelization, 2D projected depth images, 1D ordered point sequences, *etc* (Schwarz et al., 2019; Graziosi et al., 2020; Cao et al., 2021). Most PCC methods exploit 3D voxel structures and employ the octree structure to encode occupied voxels. A representative example is G-PCC (MPEG, 2025), standardized by MPEG. Another PCC method, *e.g.* MPEG V-PCC (Graziosi et al., 2020), utilizes a 3D-to-2D projection and encodes the projected depth maps, and other data using video codecs. Recently, learning-based approaches (Gao et al., 2025), such as SparsePCGC (Wang et al., 2023), leverage neural networks to compress sparse voxels, have reported superior performance. However, reconstructing meshes from point clouds often introduces severe distortions, which significantly restricts the practical applicability of these methods.

**Implicit Field-based Geometry Compression.** Implicit representations, such as signed distance functions (SDFs), can convert 3D meshes into structured tensors by uniformly sampling volumetric grids in 3D space, facilitating seamless processing with neural networks. Building on this, several prior works have attempted to compress SDFs. Tang et al. (Tang et al., 2020) employed 3D convolutional neural networks to encode TSDF voxel grids. Their method divided the original TSDF volume into non-overlapping occupied blocks of size $8 \times 8 \times 8$ and then independently compressed each block using a small autoencoder network trained in an end-to-end manner. To prevent reconstruction errors, they losslessly compressed the signs of the TSDF. NeCGS (Ren et al., 2025) introduced a learned deformation field on the TSDF volume to improve its capacity to represent fine geometric details. It constructed an auto-decoder network that was optimized on the entire dataset and included in the bitstream. However, the dense volumetric representation used in these prior methods inherently confines them to low-resolution SDF, limiting their ability to accurately capture detailed geometric structures. Although techniques like block partitioning or deformation partially alleviate the problem of representation efficiency, they incur new costs in compression: partitioning limits the use of spatial correlations, while deformation entails extra encoding overhead. These inherent trade-offs ultimately hinder efficient compression.

## 3 PROPOSED METHOD

Given a mesh model, our objective is to efficiently compress it into a compact bitstream while minimizing distortion in the decompressed models. This process consists of two key stages: **1) Geometry Representation** and **2) Data Compression**. The geometry representation stage transforms unstructured meshes into structured tensors, and the data compression stage encodes this structured data into a compact bitstream, ensuring minimal reconstruction distortion.

To this end, we propose SIR-SNC, a novel geometry compression framework. As shown in Fig. 1, it comprises two key components, a sparse implicit representation (SIR) module and a sparse neural compression (SNC) module. The SIR module efficiently converts unstructured geometry into sparse SDF tensors, from which the original surfaces can be accurately recovered. The SNC module then leverages sparse convolutional neural networks to compress these SIR tensors into the final compact bitstream, achieving high compression efficiency. Next, we will sequentially present these modules.

### 3.1 SPARSE IMPLICIT REPRESENTATION

Unlike 2D images, which consist of uniformly distributed pixels on regular grids, raw geometry data is inherently unstructured and irregular, posing challenges for direct processing with neural

networks. Recently, widely used implicit geometry representations, such as SDF, can convert the raw geometry data into regular tensors by sampling on regularly distributed grids in 3D space. However, such a dense representation requires large memory when processed, restricting their practical usage to relatively low resolutions (*e.g.*, $64^3$ or $128^3$) limiting the representation accuracy.

**Sparse SDF Tensor.** During surface extraction from implicit fields (*e.g.*, SDF) using Marching Cubes (WE, 1987), we notice that we only concentrate on the cubes near the surface while ignoring those farther away, as shown in Fig. 2. Motivated by this observation, a natural solution is to store only the grids near the surface at high resolution, ensuring detailed structural fidelity during surface extraction. Based on this, we propose sparse SDF Tensor. This sparse tensor representation enables efficient and effective representation of 3D meshes, significantly reducing memory usage and facilitating seamless integration with neural networks. At the same resolution, our sparse representation can achieve the same geometric precision as traditional SDF, while its data volume is significantly reduced by an order of magnitude. From another perspective, our method can be seen as incorporating extra surface distance information alongside discrete point positions, while providing higher precision for continuous surface geometry. Thus, it combines the efficiency of explicit discrete points and the accuracy of implicit distance fields.

Given a 3D mesh $\mathbf{S}$, we first sample regularly distributed grids throughout the space and get a set of grids $\mathbf{V}_{dense} \in \mathbb{R}^{K \times K \times K \times 3}$, where $K$ is the resolution of the grids. We then select the grids whose distance to the surface is less than a predefined threshold $\tau$ (with a default value of one voxel diagonal length), calculate their corresponding SDF according to the 3D mesh, and thereby obtain the sparse SDF tensor to implicitly represent the surface,

$$\mathbf{V} = \{(\mathbf{v}, s(\mathbf{v})) | \mathbf{v} \in \mathbf{V}_{dense}, d(\mathbf{v}) < \tau\}, \tag{1}$$

where $d(\mathbf{v})$ represents the distance from the grid point $\mathbf{v}$ to the surface, and $s$ denotes the SDF value of $\mathbf{v}$. The sign of $s(\mathbf{v})$, indicating inside or outside status, is determined via a ray casting algorithm, i.e., a ray from $\mathbf{v}$ results in $-1$ for odd intersections and $+1$ for even ones. However, when the resolution $K$ is extremely large, computing $\mathbf{V}$ becomes computationally expensive both in time and memory. To address this, we design a coarse-to-fine sampling strategy that accelerates the calculation of SDFs for the preserved grids. Further details are provided in the *Appendix*.

**Surface Extraction.** To recover the surface from the proposed sparse implicit representation, we leverage an adapted version of the Marching Cubes (WE, 1987), denoted as Sparse Marching Cubes (SMC). Specifically, for any cube whose eight vertices are present in the Sparse SDF Tensor $\mathbf{V}$, we extract the corresponding triangles following the standard Marching Cubes procedure. Compared to the traditional Marching Cubes algorithm, our method is significantly more efficient as it performs triangle extraction only in the cubes near the surface.

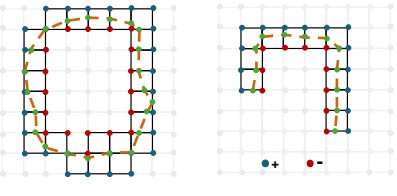

(a) Watertight    (b) Non-watertight

Figure 2: Visual illustration of the surface extraction process. The **gray** cubes are ignored in this process within our sparse representation.

***Remark.*** Since our SIR only focuses on the grids near the surface and explicitly stores their spatial positions, this localized focus enables it to robustly represent more general meshes including **non-watertight** meshes, without suffering from representation collapse. The corresponding surface extraction process is illustrated in Fig. 2b. While non-watertight surfaces lack global inside/outside consistency, the signed distances encode meaningful local geometric relationships that are sufficient to guide marching cubes in extracting the zero isosurface.

### 3.2 SPARSE NEURAL COMPRESSION

SIR effectively reduces memory and computation costs for geometry representation. To further reduce storage and transmission costs, we design a Sparse Neural Compression (SNC) module to encode it. As shown in Fig. 3, a neural network extracts compact latent features from the sparse SDF tensor via downscaling; these features are then encoded into a bitstream using entropy coding. This entire process is learned end-to-end via a compressive AutoEncoder (AE) architecture.

**Learning Latent Representation using Autoencoder.** We construct a sparse convolutional autoencoder framework to learn a compact latent feature of the original sparse SDF tensor, as shown in

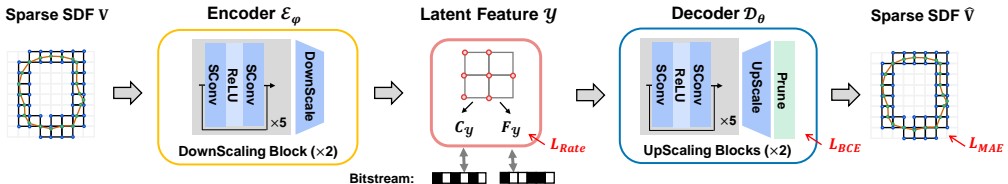

Figure 3: Sparse Neural Compression of SIR. During encoding, the Encoder $\mathcal{E}_\varphi$ downscales the original SDF tensors $\mathbf{V}$ into embedded latent features $\mathcal{Y}$, whose coordinates $C_\mathcal{Y}$ and attributes $F_\mathcal{Y}$ are separately encoded into the bitstream. During decoding, the Decoder $\mathcal{D}_\theta$ progressively reconstructs occupancy information and sparse SDFs $\hat{\mathbf{V}}$ from $\mathcal{Y}$. The Encoder and Decoder networks mainly consist of sparse convolutions (SConv). $L_{MAE}$, $L_{BCE}$, and $L_{Rate}$ denote the loss functions used in training.

Fig. 3. The encoder processes the input sparse SDF tensor through several downscaling blocks. Each block comprises stacked sparse convolution (SConv) layers with residual connections designed to fully exploit spatial correlations, followed by a voxel downscaling layer that halves spatial resolution via strided sparse convolutions. Through these downscaling blocks, features are progressively extracted and embedded into a lower-resolution sparse tensor, yielding a compact latent feature $\mathcal{Y}$ that represents the original data with fewer data. The attributes of $\mathcal{Y}$ are quantized to integers and encoded into a bitstream via entropy coding, while its coordinates are compressed separately using the G-PCC codec (MPEG, 2025).

Correspondingly, the decoder reconstructs the sparse SDF tensor from $\mathcal{Y}$ via symmetric upscaling blocks. Each block contains stacked SConv layers and a voxel upscaling layer that doubles the spatial resolution using transposed sparse convolution. To enhance computational efficiency, the decoder will prune redundant voxels after each upscaling operation: An occupancy prediction layer estimates the probability $p$ of voxel occupancy, discarding voxels below a threshold probability. This leverages the inherent sparsity of 3D data to reduce computational overhead. Finally, the sparse tensor is upsampled to the original resolution to output the reconstructed Sparse SDF tensor.

**End-to-End Rate-Distortion Optimization.** During training, we employ the Mean Absolute Error (MAE) between the reconstructed and original sparse SDF tensors as the primary loss function, i.e., $L_{\text{MAE}} = \|\mathbf{V} - \hat{\mathbf{V}}\|_1$, and use the Binary Cross-Entropy (BCE) between predicted occupancy probabilities and ground truth states as a complement, i.e, $L_{\text{BCE}}$. These two losses are combined to supervise reconstruction quality, *i.e.*,

$$L_{\text{Rec}} = L_{\text{MAE}} + \alpha \cdot L_{\text{BCE}}, \tag{2}$$

where $\alpha$ is empirically set to 0.01. To minimize the bitrate of latent features, we approximate the quantization with additive noise to preserve backpropagation differentiability, and employ a factorized entropy model (Ballé et al., 2018) to estimate the probability $p$ of latent features, from which the bitrate is calculated via the cross-entropy loss. *i.e.*, $L_{\text{Rate}} = -\sum_i \log_2(p_i)$. $L_{\text{Rate}}$ is incorporated as a constraint in the overall loss, *i.e.*,

$$L = L_{\text{Rec}} + \lambda \cdot L_{\text{Rate}}, \tag{3}$$

where $\lambda$ controls the rate-distortion trade-off.

***Remark.*** The proposed SNC module adopts a **lightweight** network architecture to meet practical complexity constraints, featuring 16-channel convolutional layers with all parameters quantized to 8-bit precision. This design enables deployment with merely around 0.42 MB in storage, while sparse computation optimizations facilitate near-real-time encoding/decoding and resource-efficient training.

Notably, the lightweight and compact network architecture of SNC further enables the efficient compression of dynamic mesh sequences by leveraging the idea of implicit neural representation (Chen et al., 2021). Specifically, it is feasible to overfit the decoder network parameters of SNC to a dynamic mesh sequence and encode the overfitted network parameters into a bitstream. In this way, we can implicitly exploit inter-frame correlations by the network parameters, avoiding complex inter-frame motion estimation on meshes.

# 4 EXPERIMENTS

In this section, we validate the effectiveness of our method through extensive experiments and comparative analyses.

## 4.1 EXPERIMENTAL SETTINGS

**Datasets.** Due to the absence of standardized neural mesh compression datasets, we follow (Ren et al., 2025) to construct a multi-source mixed dataset. The training set combines 1,000 meshes each from AMA (Vlasic et al., 2008) (human models), DT4D (Li et al., 2021) (animal models), and Thingi10K (Zhou & Jacobson, 2016) (CAD models), totaling 3,000 samples. For testing, we use: (1) a core Mixed set of 600 meshes (200 per source dataset), (2) 100 diverse CAD models from ShapeNet (Chang et al., 2015), and (3) 1,200 frames across 4 dynamic sequences in the MPEG test set (MPEG, 2023).

**Evaluation Metrics.** Following common practices in 3D reconstruction (Mescheder et al., 2019; Peng et al., 2020), We primarily use Chamfer Distance (CD) as the distortion metric, with F-Score results also reported. Additionally, we report Normal Consistency results in the *Appendix*. Compression bitrate is quantified by bitstream size. The Rate Distortion (RD) curves plot the distortion against the bitrate, with the Bjøntegaard Delta Bit Rate (BD-BR) (Bjøntegaard, 2001) measuring the relative efficiency. Encoding/decoding times (seconds) assess computational complexity.

**Methods under Comparison.** In the performance evaluation, we compare with the following representative 3D geometry compression methods, and more details of the comparison methods are provided in the *Appendix*.

- **Mesh Coding**: General-purpose *Draco* (Google, 2025) and the latest *V-DMC* standard (MPEG, 2025; Zou et al., 2025). V-DMC comparisons are confined to MPEG sequences since its Common Test Conditions (CTC) (MPEG, 2023) only provide configurations for this dataset.

- **PCC**: Standardized *G-PCC* (MPEG, 2025) and learning-based *SparsePCGC* (marked as S.PCGC) (Wang et al., 2023). We sample points from the original mesh for compression; after decoding, we reconstruct the mesh via Poisson surface reconstruction (Kazhdan & Hoppe, 2013), and remove degenerate vertices distant from the decoded point cloud for rectification.

- **SDF Compression**: The latest work in this field, *NeCGS* (Ren et al., 2025); we utilize its reported test results on the Mixed dataset and validate its performance on the ShapeNet and MPEG test sets following its open-source code. A comparison with an earlier work (Tang et al., 2020) is not feasible due to the lack of its open-source implementation and test conditions.

**Training and Variable-Rate Inference.** In the experiments, the SNC models are trained on sparse SDF tensors at a base resolution of 256. To obtain models with different rate-distortion (RD) trade-offs, we use different $\lambda$ values, *e.g.*, 0.01, 0.005, for the loss term $L_{\text{Rate}}$. Each model is optimized with Adam (Kingma & Ba, 2015) with a learning rate of 0.0001 for 50 epochs, and training is completed in one day on a single GPU. All training and testing are examined on a computer with an Intel Xeon 4309Y CPU and an NVIDIA RTX A6000 GPU.

During inference, variable-rate compression is achieved by applying a single trained model to different resolutions. Specifically, we set the resolution of the input sparse SDF tensors as $\{192, 256, 384, 512\}$ to obtain a coarse rate control. Then within the bitrate range of each resolution, fine-grained rate adjustment is achieved by selecting alternative pre-trained models corresponding to different $\lambda$ values. The proposed variable-rate method leverages the model's resolution-agnostic property to provide efficient rate adaptation without requiring model retraining. Moreover, it shows potential for achieving scalable coding, providing a direction for future research.

Table 1: BD-BR gains measured using CD for our method against the other methods.

| BD-BR Gain | Category | Verts/Faces | G-PCC | S.PCGC | Draco | V-DMC | NeCGS |
|---|---|---|---|---|---|---|---|
| **Mixed** | **mixed** | 11k/21k | -55.8% | -74.8% | -57.4% | - | -39.2% |
| **ShapeNet** | **objects** | 82k/162k | -58.0% | -91.0% | -92.0% | - | -50.8% |
| **MPEG** | **humans** | 24k/37k | -61.3% | -31.7% | -93.8% | -30.5% | -46.7% |

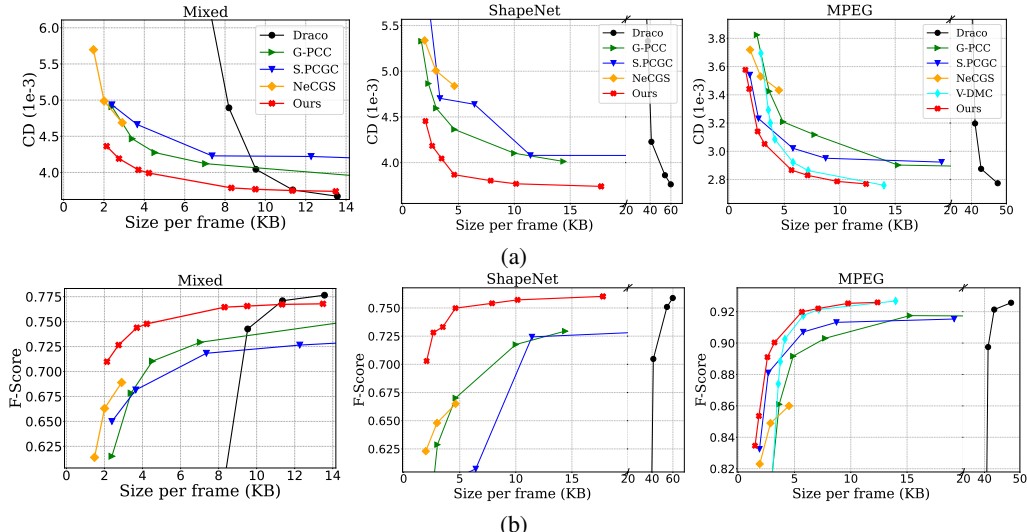

Figure 4: Quantitative comparisons of different methods on various test datasets using (a) CD, (b) F-Score as distortion metrics. Note that Draco's rate segments on ShapeNet and MPEG are plotted on a logarithmic scale for clarity, given its much higher bitrate than those of others.

## 4.2 Performance Evaluation

**Rate-Distortion Performance.** As shown in Fig. 4, our method achieves significant improvements in compression performance across a wide range of bitrates when evaluated on extensive test sets. The BD-BR values in Table 1 further quantify these bitrate savings, specifically, our method achieves substantial reductions: 50%-60% compared to G-PCC, 30%-90% compared to SparsePCGC, 50%-90% compared to Draco, 40%-50% compared to NeCGS, and 30% compared to V-DMC (on the MPEG dataset). These results demonstrate consistent superiority across various test scenarios.

In particular, our method maintains strong performance across all bitrates, while compared methods exhibit specific limitations: PCC methods, *i.e.*, G-PCC and S.PCGC, struggle to achieve low distortion in the high bitrate range. This limitation stems from their discrete point-based representations, which inherently fail to capture continuous surface details, and thus, surface reconstruction from points is constrained, with unavoidable artifacts arising in the process. Mesh compression methods, Draco and V-DMC, suffer from significant low-bitrate distortion. This is because vertex quantization and topological simplification inevitably result in noticeable geometric deformations, with Draco showing the worst performance due to simplistic strategies, Even advanced V-DMC remains constrained by this fundamental limitation. The SDF-based method NeCGS underperforms at high bitrates, mainly due to resolution constraints: its traditional dense representation caps resolution at 128, even lower than our minimum 192, directly hindering fine geometric detail capture. Beyond this, it lacks end-to-end rate-distortion optimization, further limiting compression performance.

Our method demonstrates robust generalization across diverse categories, including objects and humans, as further evidenced by the per-category comparison on the Mixed test dataset (Table 2). This robustness stems from both the mixed training dataset and the concise algorithm design. A key finding is the positive correlation between compression performance gain and mesh complexity: our method achieves more significant improvements on

Table 2: Per-Category Evaluation on the Mixed Dataset.

| Mixed | AMA | DT4D | Thingi. |
|---|---|---|---|
| **Category** | humans | animals | 3D print |
| **Verts/Faces** | 10k/10k | 18k/36k | 3.5k/7k |
| vs. Draco | -75.7% | -73.2% | -23.9% |
| vs. G-PCC | -67.0% | -73.6% | -29.1% |
| vs. S.PCGC | -35.2% | -79.2% | -57.5% |

high-fidelity meshes with abundant vertices and faces (such as those from MPEG and ShapeNet), particularly against Draco. This advantage originates from a fundamental divergence in representation. While Draco directly encodes vertices and connectivity, it is suitable for simpler models. Our approach more effectively represents and compresses complex 3D spatial structures.

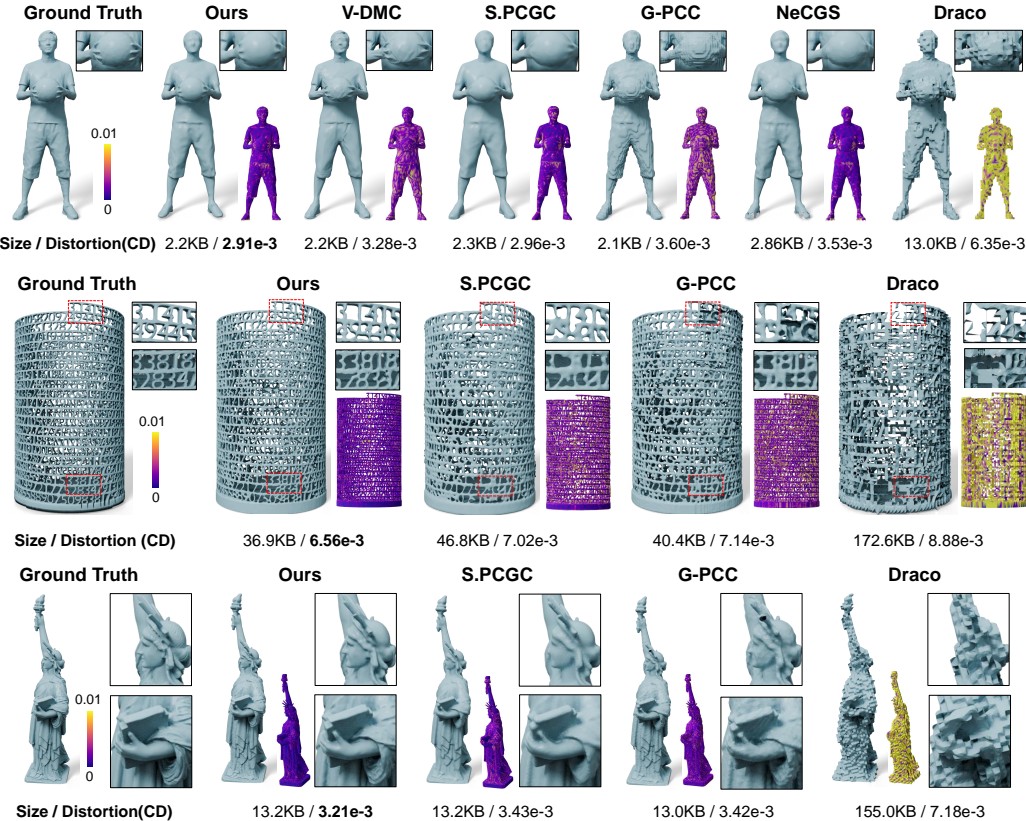

Figure 5: Visual comparisons of different compression methods on the test sets. Zoom in for details.

**Visual Comparison.** Fig. 5 visualizes the reconstructed meshes across methods; Zooming is recommended for detailed inspection. Our method consistently generates high-quality reconstructions. By contrast, PCC methods, i.e., G-PCC and SparsePCGC, struggle to capture fine surface details even at high bitrates. Draco exhibits severe block artifacts, while V-DMC shows significant deformation at low bitrates. NeCGS yields blurred details and artifacts owing to its limited resolution and redundant representation. Notably, our method tends to produce smoother surfaces with preserved geometric fidelity.

**Computational Complexity.** We evaluate the computational complexity of various methods by measuring their runtimes on the MPEG test set at medium bitrate points, as summarized in Table 3. For a comprehensive and fair comparison, we account for both the core encoding/decoding time and the pre-encoding conversion, e.g., SDF calculation, as well as the post-decoding conversion, e.g., surface reconstruction. All methods are tested on the same platform. Such a runtime comparison only gives an intuitive reference as Ours, NeCGS and SparsePCGC are prototyped using Python and run on GPU, while G-PCC and Draco are implemented using C/C++ and run on CPU.

Our method achieves highly competitive processing speeds. The core encoding/decoding times require only around 0.1s. When considering the complete pipeline, the total encoding time is slightly slower than that of Draco, while the total decoding and reconstruction time is only 0.12 s. The mesh-to-SDF conversion requires approximately 0.4 s, as it is currently implemented on the CPU, but is

Table 3: Efficiency analysis of different methods.

| Complexity | | G-PCC | S.PCGC | Draco | V-DMC | NeCGS | Ours |
|---|---|---|---|---|---|---|---|
| **Time (s)** | **Pre-Enc.** | 0.53 | 0.53 | - | - | 20 | **0.40** |
| | **Enc.** | 1.64 | 0.40 | 0.22 | 3.42 | 60 | **0.10** |
| | **Dec.** | 0.26 | 0.68 | 0.19 | 0.42 | 0.11 | **0.10** |
| | **Post-Dec.** | 5.54 | 5.22 | - | - | 0.15 | **0.02** |
| **Model Size (MB)** | | - | 6.63 | - | - | 0.77 | **0.42** |

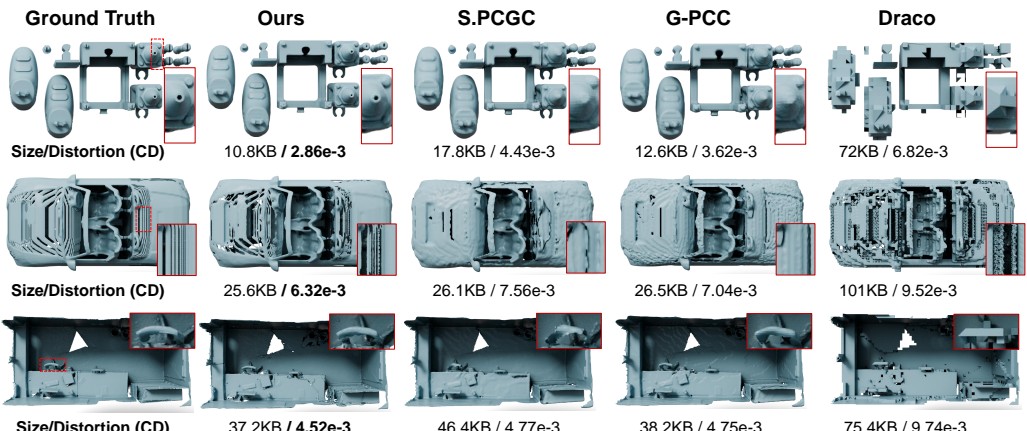

Figure 6: Reconstruction results on non-watertight surfaces. The top two rows are samples from our test set, while the bottom row is ScanNet sample. Zoom in for details.

amenable to GPU acceleration in future work. The surface extraction from SDF adds merely 0.02s. Among the comparison methods, Draco encodes/decodes in around 0.2s. V-DMC suffers from prolonged encoding that exceeds 3 s due to its complex operations. The PCC-based approaches, i.e., G-PCC and SparsePCGC, require over 5 seconds for surface reconstruction from points. The SDF-based NeCGS requires over 1 minute for encoding due to its network optimization procedure. The results underscore the computational efficiency of our approach. Our superior efficiency stems from the compact SIR and the lightweight SNC network, which is only 0.42 MB. This enables our method to be suitable for practical deployment. With further engineering optimizations, especially the migration of pre-processing to GPU, real-time performance can be readily achieved.

**Performance on Non-Watertight Surfaces.** Our method generalizes well to diverse meshes, including non-watertight surfaces. As shown in Fig. 6, we select some complex non-watertight meshes from the test dataset for validation, and conduct extended evaluations on the commonly used Scan-Net (Dai et al., 2017) datasets. In addition, we evaluated the MGN (Bhatnagar et al., 2019) open surface mesh datasets, which are shown in the *Appendix*. Notably, we used the same compression models as before without fine-tuning them on these datasets. The results confirm that our method effectively captures complex geometric features and outperforms other methods, demonstrating its robust generalization across various meshes. In theory, the intrinsic sparsity of our SIR enables a localized geometric representation restricted near the surface and governed by local geometric coherence. This, in turn, prevents reconstruction collapse that would otherwise depend on having full global topological completeness. However, while the overall reconstruction quality is preserved, minor reconstruction artifacts may appear at the boundaries of open surfaces, which is a common challenge in surface modeling. Investigating UDFs as a means of further refinement is a promising avenue for future work.

**Performance on Dynamic Mesh Sequences.** We achieve efficient compression of dynamic sequences by implicitly embedding the inter-frame shared characteristics into the overfitted decoder network parameters and encoding them into the bitstream. During training, we optimize the network parameters on the target dynamic sequence until overfitting. Consequently, during compression, the decoder parameters (*e.g.*, weights, biases) are quantized and entropy-coded alongside the per-frame latent features. Thus, the total bitrate comprises these individual latents and the shared network parameters. The proposed dynamic compression method achieves significant performance gains on the MPEG sequences. As the RD curves in Fig. 7, "Our (dynamic)" achieves around >28% bitrate reduction compared to "Ours" using feed-forward autoencoder. The de-

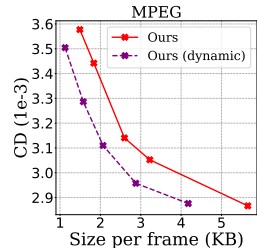

Figure 7: R-D comparison of dynamic mesh sequence compression.

coding requires around 0.1 s per frame due to the lightweight architecture, which aligns with the low-latency requirement of dynamic sequence applications. However, the encoding requires a much longer time due to the training process, making it suitable for offline encoding scenarios. It is worthwhile to investigate accelerating the encoding process in future work.

### 4.3 ABLATION STUDY

We conducted systematic ablation studies to demonstrate the effectiveness of key choices and their configurations.

**Sparsity Threshold.** In the proposed SIR, we extract grid points with distance to the surface less than a threshold $\tau$ as in Eq. 1, and organize the selected points using a sparse tensor for efficient handling. Consequently, $\tau$ determines the point quantity. As depicted in Fig. 8 left, at various resolutions, adjusting the threshold leads to a linear change in the number of points. In Fig. 8 right, we examine how $\tau$ influences distortion and note that the distortion reduces to a minimum as the threshold surpasses approximately 0.75 of

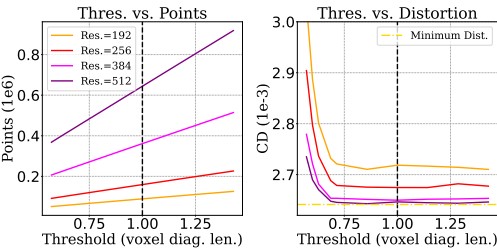

Figure 8: Ablation study of sparsity threshold.

the voxel diagonal length. For our experiments, we typically set this threshold to 1.0, which offers a sufficient buffer to maintain quality while allowing users the flexibility to adjust according to their specific needs.

**Selection of Network Hyper-parameters.** Our compression architecture is built upon stacked SConv layers to exploit spatial correlations. We primarily investigate the impact of the number of residual network blocks, as summarized in Table 4. Increasing the blocks enhances performance

Table 4: Ablation studies of residual blocks.

| # ResNets | 1 | 3 | 5 | 7 |
|---|---|---|---|---|
| **BDBR(%)** | anchor | -18% | -25% | -28% |
| **Params** | 100K | 260K | 430K | 970K |

by expanding the receptive field; based on a balance between performance and complexity, we adopt 5 ResNet blocks. In contrast, increasing the number of feature channels beyond the default 16 (e.g., to 32) significantly raises model complexity without yielding significant performance gains.

**Contribution of Occupancy Loss.** Our distortion loss combines an occupancy loss with the SDF loss (Eq. 2) to predict voxel occupancy after upscaling. This enables pruning of unoccupied voxels, maintaining sparsity throughout the decoding process. Ablation studies confirm its necessity: omitting this loss forces the retention of all voxels, which inevitably increases computational cost and introduces fragmented artifacts. In contrast, incorporating the occupancy loss improves reconstruction quality by approximately 47%. The loss weight $\alpha$ is set to 0.01 in the experiments; nevertheless, the method is robust, as values like 0.1 result in only a 2% variation.

## 5 CONCLUSION AND FUTURE WORK

We have proposed SIR-SNC, a novel framework for high-efficiency 3D mesh compression. SIR employs sparse SDF tensors to accurately represent continuous surfaces with low memory cost, while SNC utilizes a lightweight neural network to embed these sparse tensors into lower-resolution latent features, which are further compressed into bitstream via entropy coding. Our method demonstrates superior compression performance, achieving at least a 30% bitrate reduction compared to state-of-the-art mesh compression methods across diverse meshes. Additionally, and it adapts well to complex non-watertight surfaces and dynamic mesh sequences. Notably, it maintains near-real-time efficiency, with both core encoding and decoding completed in around 0.1 s using a compact 0.42 MB model, making it highly practical for deployment. Future extensions will continue to investigate efficient compression techniques for high-fidelity and large-volume meshes and incorporate texture compression capabilities.

ACKNOWLEDGMENTS

This work was supported in part by the National Natural Science Foundation of China under Grant 62422118, and in part by the Hong Kong Research Grants Council under Grants 11219324, 11219422, and N_CityU1114/25.

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

# A APPENDIX

## A.1 COARSE-TO-FINE CALCULATION OF SPARSE IMPLICIT REPRESENTATION

During the calculation of the Sparse SDF Tensor of a given mesh, we first sample uniformly distributed grids in 3D space and obtain a set of initial grids, $\mathbf{V}_{\text{init}} \in \mathbb{R}^{K_{\text{init}} \times K_{\text{init}} \times K_{\text{init}} \times 3}$, where $K_{\text{init}}$ is the initial resolution. Then we select the initial grids with a distance to the surface of less than a pre-defined threshold $\tau_{\text{init}}$, $\mathbf{V} = \{\mathbf{v}|d(\mathbf{v}) < \tau_{\text{init}}\}$. Based on $\mathbf{V}$,

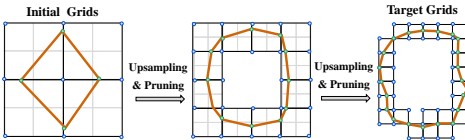

Figure 9: Coarse-to-fine Calculation of SIR.

we can obtain grids of higher resolution through upsampling, as illustrated in Fig. 9, from which we can select the grids near the surface until we achieve the required resolution. Such a sampling strategy can accelerate the calculation of Sparse SDF Tensors.

## A.2 EFFICIENCY COMPARISON OF DIFFERENT 3D REPRESENTATIONS

The memory and computational efficiency of a 3D representation are primarily determined by its sparsity, which is quantified by the number of points or voxels required. We quantitatively compare this efficiency across resolutions for various representations in Fig. 10a. TSDF volumes scale cubically with resolution, incurring substantial costs. A strategy to mitigate this is to partition the volume into $k^3$ occupied blocks (Tang et al., 2020) for independent processing. However, small block sizes (e.g., k=8) disrupt inter-block continuity and hinder the exploitation of spatial correlations vital for compression. In contrast, TSDF-Def in NeCGS(Ren et al., 2025) applies an extra deformation field to the TSDF volume to improve representation accuracy; however, this deformation relies on iterative optimization and is computationally expensive. The sparse point cloud is efficient; however, it struggles to represent surfaces with high accuracy.

Our sparse implicit representation combines the strengths of both TSDF and point cloud. It explicitly samples points near the surface, organizing them into memory-efficient sparse tensors via $[N, 3]$ coordinates, ensuring inherent sparsity and continuity. Fig. 10a shows our method requires significantly fewer points than a dense TSDF and about 4.3 times fewer than an $8^3$ blocked TSDF. Notably, this sparsity does not compromise accuracy. Compared to point clouds (Figs. 10a and 10b), our method uses only about 2.3 times as many points yet achieves a significant reduction in distortion. This optimal trade-off, summarized in Fig. 10c, underscores the superior efficiency of our representation, which effectively bridges the accuracy of TSDF with the sparsity of point clouds.

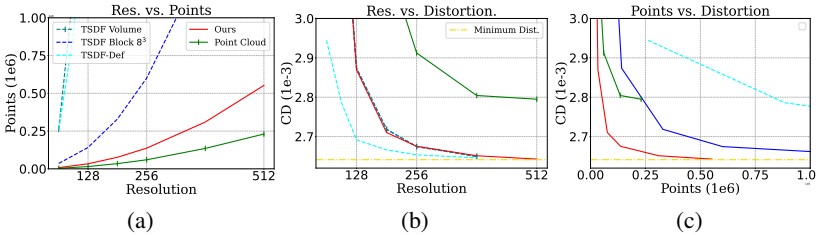

(a)        (b)        (c)

Figure 10: Comparison of efficiency and distortion across resolutions for different representations.

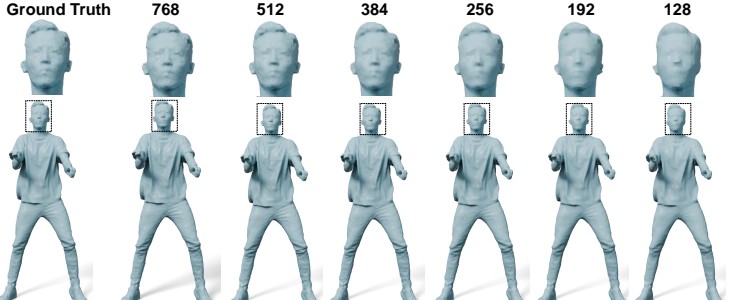

Figure 11: Visual comparison under different resolutions of SIR.

## A.3 ANALYSIS OF RECEPTIVE FIELD IMPACT ON COMPRESSION PERFORMANCE

We investigate how the receptive field influences compression performance. As indicated by the RD curve in Fig. 12, increasing the depth of the residual network to enlarge the receptive field (for instance, using a 5-resnet-block architecture instead of a single resnet block leads to a 25% improvement. While dividing the data into independent 8x8x8 blocks causes a performance drop of more than 80% even after retraining on the partition data. These results show that a large receptive field is essential for effectively modeling spatial correlations during compression. By comparison, block-based compression restricts the receptive field to small, isolated regions, which severely limits achievable performance.

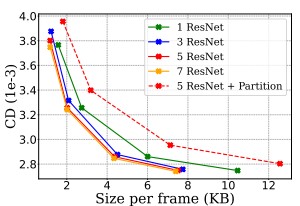

Figure 12: Ablation study of network receptive filed.

## A.4 DETAILS OF COMPARISON METHODS

For fair comparison, we used the open-source code of other methods and adhered to their respective test conditions. For PCC algorithms, we densely sampled point clouds from mesh surfaces, voxelized them to 10 bits for compression, and employed Poisson surface reconstruction with post-processing to reconstruct meshes from point clouds.

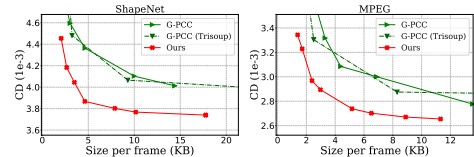

Figure 13: Comparisons with G-PCC in trisoup mode.

The configuration and bitrate settings of each method are as follows: For Draco, we adjust QP from 6 to 11 to obtain different bitrates. For V-DMC, we use all-intra mode, set QP from 22 to 44 and adjust "target" parameters to obtain optimal bitrates following its common test conditions. For G-PCC, we used octree mode and adjust positionQuantizationScale" from 0.5 to 0.125 to obtain different bitrates. Comparisons with G-PCC in trisoup mode are provided in Fig. 13. For SparsePCGC, we use its open-source pretrained models to generate various bitrates. For NeCGS, we use its reported results on the Mixed dataset and test its performance on other test sets following its open-source code.

## A.5 EVALUATION USING NORMAL CONSISTENCY

In addition to the Chamfer Distance and F-Score metrics, we also report Normal Consistency results following common practice, as shown in Fig. 14. This further demonstrates the performance of our method.

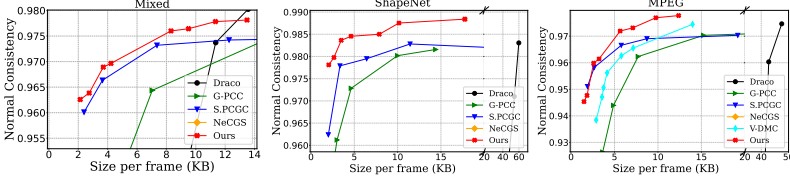

Figure 14: Quantitative comparisons of different methods on various dataset using Normal Consistency.

## A.6 MORE VISUAL RESULTS

Fig. 15 presents additional visual comparisons on the test dataset. It is evident that the reconstructed models generated by our method exhibit less distortion compared to those from the baseline methods. Fig. 16 shows results of meshes from the MGN dataset, demonstrating the effectiveness of our method in compressing open-surface meshes. Fig. 17 showcases the results of meshes with complex structures, illustrating that our method can effectively preserve detailed structures.

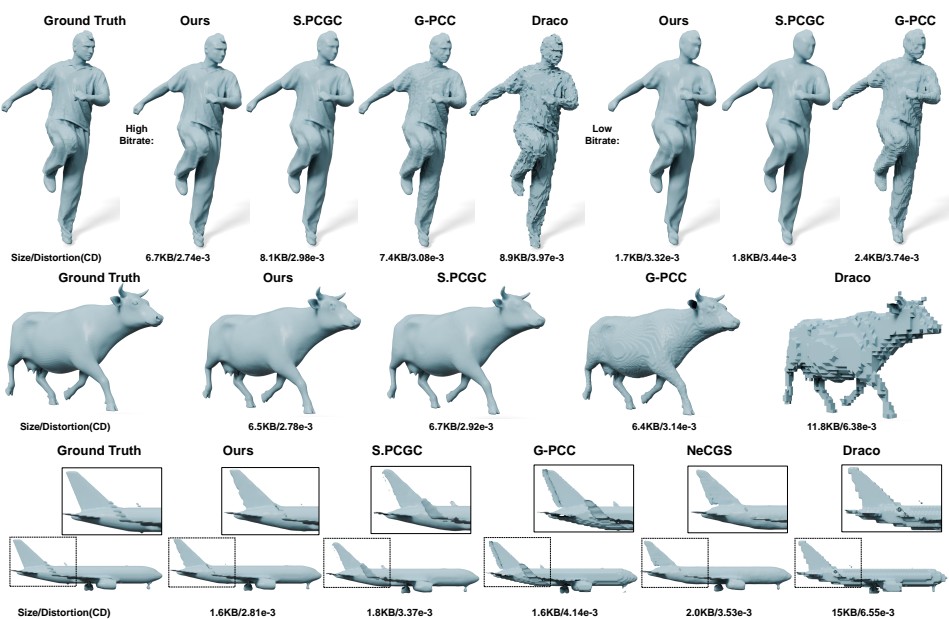

Figure 15: Visual comparisons of different compression methods. From top to bottom, the examples correspond to those from AMA, DT4D, and ShapeNet.

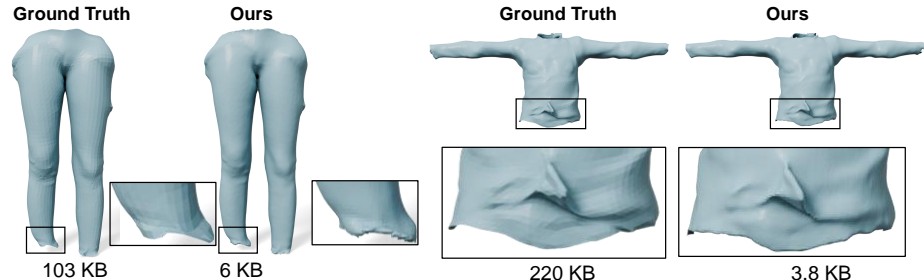

Figure 16: Visualization of decompressed results for non-watertight meshes from MGN dataset.

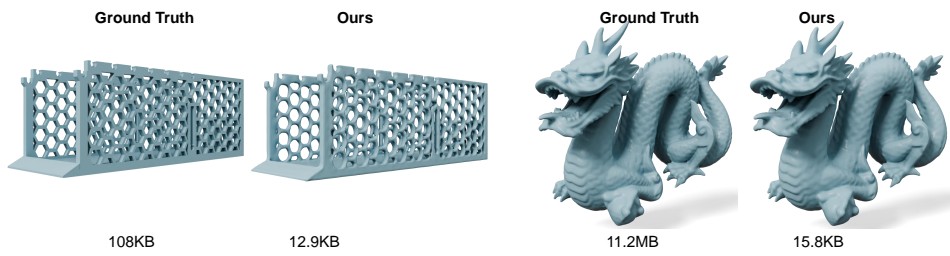

Figure 17: Visualization of decompressed results for meshes with detailed structures.

