# OpenReview forum: "Neural Compression of 3D Meshes using Sparse Implicit Representation"
_ICLR.cc/2026/Conference — ICLR 2026 Poster_

### Official Review · Reviewer_b2HU · 2025-10-28

**Soundness:** 3
**Presentation:** 4
**Contribution:** 4
**Rating:** 8
**Confidence:** 4

**Summary:**

This paper presents a novel neural mesh compression framework named SIR-SNC, which addresses the inefficiencies of existing 3D mesh compression methods. It introduces: 1) Sparse Implicit Representation (SIR): Efficiently encodes surface geometry by storing SDF values only near the surface; 2)Sparse Neural Compression (SNC): A lightweight autoencoder based on sparse convolutions that compresses the SIR representation into compact latent features and encodes them into a bitstream. Experiments on multiple datasets show that SIR-SNC significantly outperforms prior methods, including Draco, G-PCC, SparsePCGC, NeCGS, and V-DMC, both in rate-distortion performance and computational efficiency. Notably, it generalizes well to non-watertight and dynamic mesh scenarios.

**Strengths:**

1)	Technical novelty: The SIR design is a well-motivated innovation that combines the best of sparse geometry and implicit fields.
2)	Strong empirical performance and efficiency: The proposed method achieves significant BD-BR reductions across diverse benchmarks, while maintaining a lightweight model size and fast decoding speed, demonstrating both superior compression effectiveness and practical efficiency.
3)	Robust generalization: The method generalizes well to diverse and challenging meshes without any fine-tuning. It was validated on unseen datasets such as ShapeNet , ScanNet, and MGN, including non-watertight and open-surface cases, demonstrating strong robustness across domains and geometry types.

**Weaknesses:**

There is no ablation study on the predefined distance threshold.

**Questions:**

1. What are the relations between the initial grids and the target grids?
2. How sensitive is SIR to the threshold \tau?

---

> ### Author Response · Authors · 2025-11-23
> **Responses to Reviewer b2HU**
>
> We would like to thank you for your time and effort in reviewing our paper. We truly appreciate your acknowledgment of its technical novelty, strong empirical performance, and efficiency, as well as its robust generalization. In the following, we will address your concerns comprehensively.
>
>
> **Comment 1:** *There is no ablation study on the predefined distance threshold.* *How sensitive is SIR to the threshold $\tau$?*
>
>
> **Response:**
>
> In the revised manuscript (**Sec. 4.3 \"Sparsity Threshold\"**), we conduct a comprehensive ablation study to evaluate the impact of the distance threshold $\tau$ on the number of points and representation distortion. As depicted in **Fig. 8** of the revised manuscript, at various resolutions, adjusting the threshold leads to a linear change in the number of points. We also examine how $\tau$ influences distortion and note that the distortion **converges to its minimum** once the threshold exceeds approximately 0.75 of the voxel diagonal length. For our experiments, we typically set this threshold to 1.0, which offers a sufficient buffer to maintain quality while allowing users the flexibility to adjust according to their specific needs.
>
> **Comments 2:** *What are the relations between the initial grids and the target grids?*
>
> **Response:**
>
> The initial grids are presented in **Appendix A.1** of the manuscript. They serve as a **coarse, low-resolution starting point** (for example, a $2\times2\times2$ initial grid) intended to demonstrate the **coarse-to-fine computation** of the sparse SDF. In the revised manuscript, we have **added a new figure (Fig. 9)** to clarify this procedure and have renamed Appendix A.1 as "Coarse-to-fine Calculation of Sparse Implicit Representation".
>
> With an iterative process of **sparse pruning and progressive upsampling**, we preserve and further refine only those grid points
> located near the mesh surface (that is, whose distance to the surface falls below a given threshold). This process is iterated until the final sparse, high-resolution **target grids** (e.g., $512^3$) are produced. By exploiting sparsity together with progressive upsampling, the method avoids the high computational cost of direct distance calculations on a dense, high-resolution grid.

---

> ### Author Response · Authors · 2025-11-26
> **Looking forward to your further assessment**
>
> Dear Reviewer **b2HU**,
>
> Thank you for taking the time to review our manuscript and for your valuable feedback and recognition. We have carefully addressed all the comments and concerns raised, as reflected in our detailed responses and the revised manuscript.
>
> We are looking forward to your further assessment.
>
> Best regards,
>
> The Authors

---

### Official Review · Reviewer_YpR1 · 2025-11-01

**Soundness:** 3
**Presentation:** 3
**Contribution:** 3
**Rating:** 6
**Confidence:** 2

**Summary:**

This paper introduces a combination of Sparse Implicit Representation (SIR) and Sparse Neural Compression (SNC) through a two-stage approach to achieve significant file size reduction while maintaining reconstruction precision. The key insight is that by storing explicit coordinates with distance values for near-surface points only, SIR achieves high-fidelity representation with reduced file sizes compared to established methods. Results demonstrate impressive compression gains over prior methods across multiple datasets, with effective generalization to unseen meshes, support for non-watertight surfaces, and practical efficiency through sub-second decoding times.

**Strengths:**

Originality: The paper demonstrates strong originality through simple but powerful insight that only near-surface voxels matter for reconstruction, thus introducing a novel two-stage framework that stores explicit coordinates with distance values exclusively for near-surface points that achieves better compression (small resultant file size) and performance (shorter decoding time).

Quality: Sufficient comparative evidence were provided to support the authors' claims, validated across multiple datasets and benchmarked against multiple established methods. Furthermore, their cross-domain generalization testing provides convincing evidence that the method generalizes beyond its training distribution.

Clarity: The paper effectively communicated why sparse representation addresses inefficiencies in dense approaches. The primary contributions and insights are easy to understand and follow.

Significance: Substantial compression gains (30-90% bitrate reduction over state-of-the-art) across multiple datasets represents genuine practical value. The ability to handle non-watertight surfaces and performance gain could warrant additional development to transition this into a viable alternative for existing methods.

**Weaknesses:**

The absence of systematic ablation studies represents a critical methodological gap. The paper makes numerous design choices e.g., sparsity threshold, channel network width, etc., described as "empirically set" or "default" without justification. We cannot determine which components drive performance or whether results are robust across hyperparameter variations. At minimum, the authors should ablate the sparsity threshold, which dictates the rate-distortion tradeoff, network architecture choices, and loss component contributions. Without these studies, the work reads as "we tried something and it worked" rather than "we understand why our design succeeds."

Table 2 mixes heterogeneous implementations of the work (Python/GPU) and compares it against optimized C++/CPU implementations, making it hard to truly discern if the  observed speed differences originated from algorithmic insight or implementation choices.

The non-watertight surface handling lacks sufficient justification. The paper claims the method "robustly represents" non-watertight meshes based solely on showing it works in practice, without explaining why it should work reliably. While the intuition makes sense, missing the explanations of when and why the approach succeeds or fails, measurements of whether geometric properties are preserved correctly, and comparisons with methods specifically built for handling non-watertight surfaces weakens the paper. A deeper analysis showing when it works, when it doesn't, and how well it preserves geometric structure is valuable here.

**Questions:**

1. Can you provide systematic ablation studies on key design choices?
2. What is the theoretical basis for non-watertight surface handling?
3. Can you clarify the source of computational efficiency gains?
4. How does your method generalize across different mesh types and resolutions?
    - Can you provide per-category performance breakdowns showing where the method excels versus struggles?
    - What happens if the resolution exceeds $512^3$ significantly?
    - Are there mesh characteristics (topology, geometric complexity, scale) that predict compression performance?

---

> ### Author Response · Authors · 2025-11-23
> **Responses to Reviewer YpR1 (1/3)**
>
> We would like to thank you for your time and effort in reviewing our paper. We truly appreciate your acknowledgment of our strong originality and significant compression performance. Thank you for your constructive comments. In the following, we will address your concerns comprehensively.
>
> **Comment 1:** *The absence of systematic ablation studies represents a critical methodological gap. \... At minimum, the authors should ablate the sparsity threshold, which dictates the rate-distortion tradeoff, network architecture choices, and loss component contributions. \... Can you provide systematic ablation studies on key design choices?*
>
> **Response:**
>
> Thank you for the suggestions. We have added systematic ablation studies in **Sec. 4.3 of the revised manuscript** to evaluate the sparsity threshold, network architecture, and loss components. The main updates are summarized below:
>
> -   **Sparsity Threshold:** We experimentally assessed the impact of the sparsity threshold on representation efficiency and reconstruction distortion: **1)** Adjusting the threshold leads to a linear change in the number of points; **2)** Distortion converges to its minimum as the threshold surpasses approximately 0.75 of the voxel diagonal length. we typically set this threshold to 1.0 to provide a sufficient buffer to maintain quality while allowing users the flexibility to adjust according to their specific needs.
>
> -   **Network Architecture:** We mainly examine how varying the number of residual blocks affects performance. Increasing the number of residual network blocks enlarges the receptive field and leads to better results; for instance, raising the count from 1 block to 5 blocks yields roughly a 25% improvement. While increasing the number of feature channels beyond the default of 16 to 32 substantially boosts model complexity; however, it does not lead to notable improvements in performance.
>
> -   **Loss Components:** We empirically assessed the impact of the occupancy loss: when it is removed, all voxels must be retained, leading to higher computational cost and causing fragmented artifacts. In contrast, including the occupancy loss enhances
>     reconstruction quality by about 47%. In our experiments, the loss weight $\alpha$ is set to 0.01; however, the approach remains
>     stable, as using values like 0.1 changes performance by only around 2%.
>
> **Comment 2**: *Table 2 mixes heterogeneous implementations of the work (Python/GPU) and compares it against optimized C++/CPU implementations, making it hard to truly discern if the observed speed differences originated from algorithmic insight or implementation choices.*
>
> **Response:**
>
> The main source of efficiency in our approach stems from our algorithmic advances, most notably the introduced **sparse representation**, rather than from specific implementation optimizations. This becomes clear when compared with other learning-based approaches, such as SparsePCGC and NeCGS, which were evaluated on the same GPU hardware.
>
> **Efficient Sparse Representation:** To quantitatively demonstrate the efficiency of our representation against alternatives, we compare the number of points and distortion for different representations in **Sec. A.2 of the revised Appendix**. To provide an intuitive comparison, we present an example in the table below. Clearly, compared to the point cloud representation used by SparsePCGC and the deformable TSDF volume representation used by NeCGS, our method achieves significantly lower
> distortion while using fewer points. Since computational complexity scales linearly with the number of points, our superior sparsity
> directly reduces the associated overhead.
>
> | **Representations**       | **Resolution** | **Points** | **Dist.(CD)** |
> | ------------------------- | -------------- | ---------- | ------------- |
> | **Ours**                  | 192            | **76k**    | **2.71e-3**   |
> | **Point Cloud** (S.PCGC)  | 512            | 230k       | 2.79e-3       |
> | **TSDF-Def** (NeCGS)      | 96             | 884k       | 2.79e-3       |
>
> **Lightweight and Feed-forward Compression Network:** Our compression network is built with efficiency in mind. It has fewer parameters (e.g., 0.42MB compared to SparsePCGC's 6.63MB) and relies on a feed-forward autoencoder, thereby avoiding the iterative optimization strategy of NeCGS, which leads to lengthy encoding times.
>
> **The Effects of Variations in Implementation:** When compared to **traditional methods**, e.g., Draco, VDMC, and G-PCC, which rely on C++/CPU implementations, the runtime comparison serves only as an **intuitive reference** for complexity comparison. Our neural network--based approach is inherently well-suited for GPU processing. When run on a CPU, the encoding time rises from 0.1 seconds to about 3 seconds. This runtime gap underscores the substantial speedup our method can achieve, especially given that the current implementation relies solely on standard PyTorch without any low-level optimizations.

---

> ### Author Response · Authors · 2025-11-23
> **Responses to Reviewer YpR1 (2/3)**
>
> **Comment 3:** *The non-watertight surface handling lacks sufficient justification. \... While the intuition makes sense, missing the explanations of when and why the approach succeeds or fails, measurements of whether geometric properties are preserved correctly, and comparisons with methods specifically built for handling non-watertight surfaces weakens the paper. A deeper analysis showing when it works, when it doesn't, and how well it preserves geometric structure is valuable here. What is the theoretical basis for non-watertight surface handling?*
>
> **Response:**
>
> **Theoretical Foundation:** Our sparse SDF representation handles non-watertight surfaces through a **\"locality principle\"**: surface reconstruction depends on local geometric coherence, not global topological completeness. By processing only surface-proximal points, it decouples surface reconstruction from global topology. Specifically,
> this enables our approach to **spatially separate** geometrically valid regions from areas with topological issues, **assign signs locally** via ray-casting, and **reconstruct surface patches independently**, each relying solely on local coherence. This is theoretically justified, since surface reconstruction is fundamentally a local geometric problem, not a global topological one.
>
> **Experimental Validation:** **Fig. 6** in the manuscript presents intuitive visual comparisons of reconstruction results on non-watertight surfaces, together with the objective metric, i.e., Chamfer distance. We benchmark our method against widely used general-purpose techniques, e.g., Draco and G-PCC. The results consistently demonstrate our method's capability in handling non-watertight meshes.
>
> As a lossy compression technique, our method preserves the overall reconstruction quality and can reliably recover most regions of non-watertight meshes. However, minor reconstruction artifacts may appear in some complex cases; for example, **at the boundaries of open surfaces** where the sign is ambiguous, as shown in the car example in Fig. 6 and the pant example in Fig.16 of the manuscript. Accurately reconstructing these ambiguous boundaries remains a common difficulty in
> surface modeling.
>
> Future work could explore enhanced representations; for instance, by integrating UDFs and applying dedicated post-processing techniques to better handle these artifacts and further improve performance.

---

> ### Author Response · Authors · 2025-11-23
> **Responses to Reviewer YpR1 (3/3)**
>
> **Comment 4:** *How does your method generalize across different mesh types and resolutions?*
>
> *a) Can you provide per-category performance breakdowns showing where the method excels versus struggles?*
>
> *b) What happens if the resolution exceeds $512^3$ significantly?*
>
> *c) Are there mesh characteristics (topology, geometric complexity, scale) that predict compression performance?*
>
> **Response:**
>
> **a) Generalization Across Mesh Types.**
>
> To better analyze the performance across different meshes, we have included the category information (e.g., human, objects) and mesh complexity (quantified by the number of vertices/faces) in **Table 1** in the revised manuscript. In addition, we provide a per-category evaluation of the Mixed test dataset in **Table 2**. For convenience, we
> copy them below:
>
> | **BD-BR Gain** | **Category** | **Verts/Faces** | **G-PCC** | **S.PCGC** | **Draco** | **V-DMC** | **NeCGS** |
> |----------------|--------------|-----------------|-----------|------------|-----------|-----------|-----------|
> | **Mixed**      | **mixed**    | 11k/21k         | -55.8%    | -74.8%     | -57.4%    |   -   | -39.2%    |
> | **ShapeNet**   | **objects**  | 82k/162k        | -58.0%    | -91.0%     | -92.0%    |   -   | -50.8%    |
> | **MPEG**       | **humans**   | 24k/37k         | -61.3%    | -31.7%     | -93.8%    | -30.5%    | -46.7%    |
>
>
> | **Mixed** | **AMA** | **DT4D** | **Thingi10K** |
> |-------------|-----------|---------|----------|
> | **Category**    | humans    | animals | 3D print |
> | **Verts/Faces** | 10k/10k  | 18k/36k | 3.5k/7k  |
> | vs. Draco   | -75.7%    | -73.2%  | -23.9%   |
> | vs. G-PCC   | -67.0%    | -73.6%  | -29.1%   |
> | vs. S.PCGC  | -35.2%    | -79.2%  | -57.5%   |
>
> Our method exhibits **strong generalization across a wide range of content**, including objects, humans, and animal meshes with diverse geometric characteristics. This is clearly reflected by the quantitative BD-BR comparisons in Tables 1 and 2, as well as in the visual comparisons in **Fig. 5** and the additional visual results in **Fig. 15** of the manuscript. This robustness stems from both the mixed training dataset and the concise algorithm design.
>
> A key finding is the positive correlation between compression performance gain and mesh complexity: our method achieves more **significant improvements on complex meshes with more vertices and faces** (such as those from MPEG and ShapeNet), particularly when compared to Draco. This advantage originates from a fundamental divergence in representation. While Draco directly encodes vertices and connectivity, it is suitable for simpler models with lower vertex
> counts. Our approach more effectively represents and compresses complex spatial structures.
>
> **b) Generalization Across Resolutions.**
>
> Our method is **resolution-agnostic** and inherently supports a wide range of resolutions. In our experiments, we train a single compression model at a base resolution of 256 and then apply it to multiple resolutions during inference to realize **variable-rate compression**, showing robust generalization across different resolutions. Specifically, we set resolutions of $\{192, 256, 384, 512\}$. this choice is based on the observation that, for most models, a resolution of up to 512 is sufficient to achieve an accurate representation without the necessity for higher values. As illustrated in **Fig. 10** of the revised manuscript and the **table below**, using a resolution of 512 nearly achieves the minimum Chamfer Distance (CD) distortion when the ground-truth meshes are compared directly. In practical applications, users can freely choose the resolution according to their specific needs, allowing them to achieve the best trade-off between accuracy and complexity.
>
> | **Resolution** | 192 | 256 | 384 | 512 | 768 | 1024 | lossless |
> | -------------| --| --| --| -- | --| ---| :------- |
> | **Points**     | 76k | 137k | 310k | 552k | 1246k | 2217k | - |
> | **CD (1e-3)**  | 2.710 | 2.675 | 2.651 | 2.642 | 2.641 | 2.641 | 2.641 |
>
> **c) The Effect of Mesh Complexity on Compression Efficiency.**
>
> As in our response to "Generalization Across Mesh Types," we have now included the **number of vertices/faces** to reflect mesh complexity in the revised manuscript. We observe that our method delivers **larger gains on more complex meshes** with higher vertex and face counts (such as those from MPEG and ShapeNet), especially in comparison to Draco. This benefit stems from a fundamental difference in how the data is represented. Draco explicitly encodes vertices and connectivity, which is suitable for simpler models with fewer vertices. In contrast, our method is better suited to capture and compress intricate spatial structures.

---

> ### Author Response · Authors · 2025-11-26
>
> Dear Reviewer **YpR1**,
>
> Thank you for taking the time to review our manuscript and for your valuable feedback and recognition. We have carefully addressed all the comments and concerns raised, as reflected in our detailed responses and the revised manuscript.
>
> We are looking forward to your further assessment.
>
> Best regards,
>
> The Authors

---

### Official Review · Reviewer_EZUy · 2025-11-01

**Soundness:** 2
**Presentation:** 3
**Contribution:** 2
**Rating:** 4
**Confidence:** 4

**Summary:**

This paper proposes a framework for 3D mesh compression called SIR-SNC.The paper introduces a two-stage approach, 1) Sparse Implicit Representation (SIR): A mesh representation that stores Signed Distance Field (SDF) values only on a regular grid of points near the object's surface, 2) Sparse Neural Compression (SNC): A lightweight, sparse convolutional autoencoder (AE) designed to compress the SIR. A Sparse Marching Cubes (SMC) algorithm is used for efficient surface extraction from the SIR.

**Strengths:**

- Promising results on the evaluated benchmarks. The paper shows large gains on most of the benchmarks.
- Efficiency of method: The proposed method appears to be quite practical using a small model and fast encoding/decoding.
- The method works on general non-watertight and open meshes.

**Weaknesses:**

- Limited Novelty: The proposed approach appears to closely follow that of Tang et al., without sufficiently acknowledging the fact and without presenting relevant empirical comparison. It is not clear what aspects of the proposed approach are truly original. More on this in the next point.
- Insufficient comparison with Tang et al.: The method seems to be heavily inspired by that of Tang et al (cited in the paper). As such, it is not clear what aspects of the paper are novel and what are borrowed from existing works. For example, Tang et al, also uses a sparse representation, using an occupancy map (which is losslessly coded and transmitted) to encode the occupied voxels. These occupied voxels are also found using marching cubes algorithm (like the current paper). This allows Tang et al. to avoid coding and transmitting voxels that don’t carry useful information (Fig 3 in Tang et al.). This fact is not mentioned in the paper and/or attributed to Tang et al, though the figures of Sparse SDF in the paper (e.g. Figure 4, first and last diagram) are reminiscent of Figure 3 in Tang et al. Similarly, Tang et al, uses a small autoencoder to encode/decode features of only the occupied blocks of voxels. Again, this is not acknowledged/mentioned in the paper. Further, Tang et al, also uses a rate-distortion loss (eq 1 in Tang et al) to train a compressed representation (similar to eq. 3 in current paper).

Overall, I find it quite concerning that sufficient credit/comparison is to made with the prior art of Tang et al, despite the obvious similarities in the proposed method. While the presented results are promising, I am quite curious about the source of this omission.

**Questions:**

Please elaborate extensively on the differences from the method of Tang et al. and exactly what aspects of the proposed method are claimed to be novel and validated via the experiments.

---

> ### Author Response · Authors · 2025-11-23
> **Responses to Reviewer EZUy (1/2)**
>
> Thank you for your time and effort in reviewing our manuscript. We appreciate your acknowledgment of our strengths in promising results, efficiency, and generalization.
>
> However, we **disagree** with your questioning of our **originality** in comparison to the prior work of Tang et al (Tang, Danhang, et al. \"Deep implicit volume compression.\" CVPR. 2020). The core contribution of our work lies in the introduction of a **sparse tensor-based implicit representation** and the corresponding compression framework. This is fundamentally different from the **block-based volumetric representation** and block-based coding used by Tang et al.
>
> Next, in response to your comments, we will clarify the differences between our work and Tang et al. in detail.
>
> **Comment 1:** *Tang et al also use a sparse representation, utilizing an occupancy map (which is losslessly coded and transmitted) to encode the occupied voxels. \... This allows Tang et al. to avoid coding and transmitting voxels that do not carry useful information (Fig 3 in Tang et al.). This fact is not mentioned in the paper and/or attributed to Tang et al, though the figures of Sparse SDF in the paper (e.g., Figure 4, first and last diagrams) are reminiscent of Figure 3 in Tang et al.*
>
> **Response:**
>
> We believe there is a **misunderstanding** regarding the fundamental difference between our **sparse tensor representation** and the **block-based volumetric representation** employed by Tang et al. Their method divides the TSDF volume into non-overlapping occupied 8x8x8 blocks, with **each block compressed independently**. (This fact is described in the second paragraph of Sec. 4 of Tang et al: \"*\..., we process each frame independently in a block based manner. From the TSDF volume V, we extract all non-overlapping blocks of size $k\times k\times k$ that contain a zero crossing. We refer to these blocks as occupied blocks and compress them independently.*) This differs fundamentally from our sparse representation: Tang et al. represent a mesh using multiple independent blocks of size $\mathbf{k\times k\times k}$, whereas we represent a mesh as a sparse tensor characterized by coordinates of size $\mathbf{N\times3}$ and associated SDF attributes sized $\mathbf{N\times1}$.
>
> The block-based coding method employed by Tang et al. requires **compressing all voxels within a block**, which inevitably includes numerous unoccupied voxels. Thus, their approach does not achieve the objective of \"avoiding coding and transmitting voxels that don't carry useful information\" in the manner that our method does. (A detailed overview of their block compression pipeline is provided in Fig. 11 of the supplementary material of Tang et al.)
>
> There is also a fundamental difference between the mentioned Fig. 3 in Tang et al. and our Fig. 4. The objective of Tang et al. is to
> losslessly compress the **signs of all voxels within a block** to minimize inference error (The description can be found in the caption of Fig. 3 and the first paragraph of Sec. 4.1 of Tang et al.), whereas our method encodes solely the **coordinates of occupied voxels**, thereby maintaining a sparse representation.
>
> More importantly, the block-based coding approach suffers from a significant limitation: processing each block independently limits the receptive field to a single block and **hinders the use of spatial correlations across blocks**. (The authors themselves acknowledged this limitation in the last paragraph of their Sec. 7: *\"In our architecture, we have assumed blocks to be i.i.d., and dropping this assumption could further increase the compression rate\"*.)
>
> In contrast, our proposed SIR employs a **sparse tensor-based representation**. Rather than processing dense volumetric blocks, it
> directly represents a 3D mesh using only the sparse set of surface-proximal points, organized as a sparse tensor with coordinates
> **\[N, 3\]**. This constitutes not just a more sparse data structure but **an entirely different representation that allows for the direct
> processing of an entire 3D space** using sparse tensor or point-based networks.

---

> ### Author Response · Authors · 2025-11-23
> **Responses to Reviewer EZUy (2/2)**
>
> **Comment 2:** *Similarly, Tang et al. uses a small autoencoder to encode and decode features of only the occupied blocks of voxels. Again, this is not acknowledged/mentioned in the paper.*
>
> **Response :**
>
> The autoencoder-based compression framework is a well-established architecture, widely adopted in many studies. Our autoencoder is not based on the design of Tang et al; instead, it is an original and standalone design specifically crafted for our sparse representation. It differs fundamentally in its **data format**, **network structures**, and **model capacity**.
>
> Tang et al's autoencoder is utilized locally within a predetermined **$8 \times8 \times8$ block**, as previously mentioned. According to the first paragraph of Sec. 9 in Tang et al, it is \"implemented with **three convolutions** in the encoder and three transposed convolutions in the decoder\".
>
> In contrast, our autoencoder handles the entire sparse representation, which can span a large range of **arbitrary resolutions** up to $512^3$ and higher, and directly transforms it into a lower-resolution sparse tensor with embedded latent features. Built on **deep residual network blocks**, our model achieves a **large receptive field**, thereby capturing extensive spatial correlations for better compression.
>
> **Comment 3:** *Further, Tang et al, also uses a rate-distortion loss (eq 1 in Tang et al) to train a compressed representation (similar to eq. 3 in current paper).*
>
> **Response:**
>
> The rate-distortion loss function is a standard component across various lossy compression frameworks.
>
> In Tang et al, the loss function is designed for their volumetric block representation, which incorporates sophisticated weighting and masking techniques to **handle the many empty voxels in each block and to highlight occupied areas**. (This detail can be found in Sec. 4.2 of Tang et al.)
>
> In contrast, our sparse representation naturally **eliminates unoccupied voxels through the pruning operation**. As a result, our RD loss (refer to Eq. 2 in the manuscript) becomes much simpler: it utilizes a straightforward combination of the MAE loss for the SDF value and the BCE loss for occupancy, which is more efficient and robust.
>
> **Comment 4:** *Please elaborate extensively on the differences from the method of Tang et al. and exactly what aspects of the proposed method are claimed to be novel and validated via the experiments.*
>
> **Response:**
>
> The main innovation of our study lies in its **sparse tensor-based representation and compression framework** for SDFs, which marks a significant shift from previous volumetric techniques. We have expanded the discussion in Sec. 2 \"Related Work\" to clarify these distinctions, providing a detailed comparison with prior SDF compression methods that outlines our method's fundamental advancements.
>
> Regarding the empirical comparison with Tang et al, we have made efforts to conduct a direct comparison. However, to the best of our knowledge, there is **no publicly available implementation** of their approach, and the datasets they employed are **proprietary and inaccessible**. We contacted the authors via email during the rebuttal period but, regrettably, did not receive a reply. Given these constraints in replicating their results, a direct comparison of compression performance is not feasible.
>
> To demonstrate the significance of our proposed innovations and clarify how they differ from previous work, we conducted extensive experiments in the revised manuscript:
>
> -   In A.2 of the revised Appendix, We compare the efficiency of our     **sparse representation** to that of the volumetric block     representation, showing an approximate **74%** reduction in the  **number of voxels**.
>
> -   In Sec. 4.3 and A.3, ablation studies on the network architectures     showed a **25%** BD-BR gain when using a **deep network** instead of  a shallow one, and more than an **80%** BD-BR degradation when **block partitioning** is enabled.
>
> -   In Sec 4.3, ablation experiments on the loss function reveal a **47%** performance improvement by adding an additional **pruning operation** combined with a simple **occupancy loss**.

---

> ### Author Response · Authors · 2025-11-26
> **Looking forward to your further assessment**
>
> Dear Reviewer **EZUy**,
>
> Thank you for taking the time to review our manuscript and for your feedback. We have carefully addressed all the comments and concerns raised, as reflected in our detailed responses and the revised manuscript.
>
> We are looking forward to your further assessment.
>
> Best regards,
>
> The Authors

---

> ### Comment · Reviewer_EZUy · 2025-11-26
> **Acknowledgement**
>
> I have read the other reviews and corresponding author rebuttals. In my opinion, the proposed approach appears to be quite similar to that of Tang et al. at a high level, but differs sufficiently in details and lower level choices (e.g. block of voxels near surface vs sparse set of points near surface etc.).
>
> Overall, authors have sufficiently addressed most of my concerns and I will be increasing my rating.

---

> > ### Author Response · Authors · 2025-11-26
> > **Responses to Reviewer EZUy**
> >
> > We are delighted to receive your positive assessment and the acknowledgment of the sufficient differences in our technical approach compared to Tang et al. We sincerely appreciate your favorable recommendation and the decision to raise the rating for our work.

---

### Author Response · Authors · 2025-11-22
**Official Comment by Authors**

We sincerely appreciate the reviewers for dedicating their valuable time and effort to assess our work and acknowledge its merits. We are
thankful to all the reviewers for acknowledging the strong performance and efficiency of our approach, and especially to reviewers YpR1 and b2HU for highlighting the novelty of our method.

In our updated manuscript, we have carefully addressed the issues raised by the reviewers. The main modifications, highlighted in
[**red**] within the revised PDF, are summarized below:

-   **Sec. 2 \"Related Work\":** We have expanded the discussion of  prior work on SDF compression, emphasizing the main distinctions
    between their dense volume representations and our sparse tensor  representation, which constitutes the core contribution of our
    method.

-   **Sec. 4.2 \"Performance Evaluation\":** We have expanded our analysis to better illustrate our method's generalization capability
    across diverse meshes and to emphasize its strengths in compressing complex meshes. We also provided a more thorough justification for non-watertight surface handling.

-   **Sec. 4.3 \"Ablation Study\":** We conducted systematic ablation studies to validate the impact of key design decisions and their
    configurations, including the sparsity threshold, the choices of network architecture, and the contribution of the occupancy loss.

-   **Appendix:** In A.2, we provided more comprehensive and quantitative evaluations of the efficiency of various 3D representations, with particular emphasis on the SDF-based methods covered in the \"related work\" section. In A.3, we reported experimental results demonstrating how different receptive field sizes in the compression method influence overall compression performance.

We appreciate the reviewers' insightful comments, which have significantly helped us refine and strengthen our work.

---

> ### Author Response · Authors · 2025-11-25
> **Looking forward to your further comments**
>
> Dear **Reviewers**,
>
> Thank you for taking the time to review our manuscript and for your valuable feedback and recognition. We have carefully addressed all the comments and concerns raised, as reflected in our detailed responses and the revised manuscript.
>
> We sincerely appreciate your efforts and look forward to your further assessment.
>
> Best regards,
>
> The Authors

---

### Comment · Area_Chair_5ibY · 2025-11-27

Dear Reviewers,

Author responses are now posted. Please add your discussion comment(s) and update score/confidence as needed. Thank you!

Best regards,

AC

---

### Author Response · Authors · 2025-11-30
**Rebuttal Summary for the Area Chair**

Dear Area Chair,

We kindly summarize the reviewers' comments and our responses from the rebuttal phase, hoping to assist you in efficiently evaluating this process. We sincerely thank the reviewers for their time and constructive feedback. Throughout the initial reviews and subsequent discussions, We are pleased that​ **the reviewers consistently recognized the strong performance and originality** of our *Sparse Implicit Representation (SIR)* for mesh compression.


The main concerns raised during the review process were:

-   **The novelty of our approach** compared to that of Tang et al. (Reviewer EZUy).

-   **The need for ablation studies** on key design choices, including  the sparsity threshold (Reviewers YpR1, b2HU).

-   **Further justification** regarding non-watertight surface handling, computational efficiency, and generalization ability (Reviewer YpR1).

We have carefully addressed these concerns in our responses and the revised manuscript as follows:

-   **Novelty clarification compared to Tang et al:** We elaborated on the fundamental difference between our *sparse tensor-based representation* and Tang et al.'s *block-based volumetric representation*, underscoring SIR's **more compact representation** and its capability for **direct processing of 3D space without requiring partitioning**.

-   **Systematic ablation studies:** We have added extensive ablations (Sec. 4.3) on the sparsity threshold, network architecture, and loss components, **quantitatively evaluating the impact of key design choices**.

-   **Enhanced performance justification:** We provided (i) a **theoretical basis** for non-watertight surface handling; (ii) a **detailed analysis** of the computational efficiency gains; and (iii) an in-depth justification of **strong generalization** across diverse mesh types and resolutions.

We are encouraged by the positive feedback from the discussion: **Reviewer EZUy** stated that we had \"sufficiently addressed most of my concerns,\" **acknowledged the sufficient differences from Tang et al.,
and raised the rating to 6**. Additionally, **reviewers YpR1 and b2HU** raised no further concerns after our responses, **maintaining their positive ratings**.

In summary, we have thoroughly addressed all key concerns raised by the reviewers and improved our manuscript. We sincerely appreciate the reviewers for their insightful comments and the Area Chair for the
effort and time.

Best regards,

The Authors

---

### Meta-Review · Area_Chair_Quyu · 2026-01-06

**Summary:**

The paper proposes a novel neural mesh compression framework called SIR-SNC, which aims to address inefficiencies in existing mesh representations and demonstrates SoTA performance.

**Strengths**
- Technical Novelty: The reviewers acknowledged the originality of the design
- Strong Performance: The method demonstrates significant bitrate reductions (30-90% over SOTA) and noticeable computational efficiency across different benchmarks.
- Generalization: Also generalize well to unseen datasets like ScanNet and handle open surfaces

**Weakness and Rebuttal Resolution**
During the review phase, reviewers raised three primary concerns, all of which were satisfactorily addressed by the authors during the rebuttal.
- Novelty over Tang et al. (CVPR'20): the authors emphasize the sparse tensor representation without partitioning over their volumetric blocks. The reviewer EZUy acknowledged it and raised their rating
- Ablation: Multiple reviewers requested additional ablations for design choices like sparsity threshold. The authors provided these studies.
- Non-watertight surfaces: questions regardin the theoretical basis for handling open surfaces were raised. The authors provided justification based on the locality principle.

The authors were highly responsive during the rebuttal and I believe they have addressed the key concerns with new experiments and clarfiications. All reviewers support acceptance (score 8, 6, and and a committed increase from 4). Therefore, I agree the work meets the bar for ICLR and would like to accept it.

**Reviewer Concerns:**

I believe most concerns have been resolved.

**Reviewer Scores:**

**YpR1**: probably will keep the positive rating or increase to 8

**b2HU**: already gave 8 before the rebuttal period, so probably it will keep it since the authors have resolved their concerns

**EZUy**: promised to increase from 4, so I believe it can be increased at least to 6.

---

### Decision · Program_Chairs · 2026-01-26

Accept (Poster)